# Impaired Function of Solute Carrier Family 19 Leads to Low Folate Levels and Lipid Droplet Accumulation in Hepatocytes

**DOI:** 10.3390/biomedicines11020337

**Published:** 2023-01-31

**Authors:** Ainara Cano, Mercedes Vazquez-Chantada, Javier Conde-Vancells, Aintzane Gonzalez-Lahera, David Mosen-Ansorena, Francisco J. Blanco, Karine Clément, Judith Aron-Wisnewsky, Albert Tran, Philippe Gual, Carmelo García-Monzón, Joan Caballería, Azucena Castro, María Luz Martínez-Chantar, José M. Mato, Huiping Zhu, Richard H. Finnell, Ana M. Aransay

**Affiliations:** 1Food Research, AZTI, Basque Research and Technology Alliance (BRTA), Parque Tecnologico de Bizkaia, Astondo Bidea, Building 609, 48160 Derio, Spain; 2OWL Metabolomics, Parque Tecnologico de Bizkaia, Building 502, 48160 Derio, Spain; 3Department of Nutritional Sciences, Dell Paediatric Research Institute, The University of Texas at Austin, Austin, TX 78712, USA; 4CIC bioGUNE, Parque Tecnologico de Bizkaia, Building 801-A, 48160 Derio, Spain; 5CIBERehd, ISCIII, 28029 Madrid, Spain; 6Ikerbasque, Basque Foundation for Science, 48009 Bilbao, Spain; 7Nutriomics Research Group, Nutrition Department, Pitié-Salpétrière Hospital, INSERM, Sorbonne Université, F-75013 Paris, France; 8INSERM, UMR_S 1166, NutriOmics Team 6, F-75013 Paris, France; 9Assistance Publique Hôpitaux de Paris, Nutrition department ICAN and CRNH-Ile de France, Pitié-Salpêtrière Hospital, F-75013 Paris, France; 10Team 8 “Chronic Liver Diseases Associated with Obesity and Alcohol”, INSERM, U1065, Centre Hospitalier Universitaire de Nice, C3M, Université Côte d’Azur, 06000 Nice, France; 11Liver Research Unit, Santa Cristina University Hospital, Instituto de Investigación Sanitaria Princesa, 28009 Madrid, Spain; 12Liver Unit, Hospital Clinic, 08036 Barcelona, Spain

**Keywords:** NAFLD, MAFLD, folate cycle, liver lipid metabolism, SNPs, genomics, transcriptomics, metabolomics

## Abstract

Low serum folate levels are inversely related to metabolic associated fatty liver disease (MAFLD). The role of the folate transporter gene (*SLC19A1*) was assessed to clarify its involvement in lipid accumulation during the onset of MAFLD in humans and in liver cells by genomic, transcriptomic, and metabolomic techniques. Genotypes of 3 SNPs in a case-control cohort were initially correlated to clinical and serum MAFLD markers. Subsequently, the expression of 84 key genes in response to the loss of *SLC19A1* was evaluated with the aid of an RT^2^ profiler-array. After shRNA-silencing of *SLC19A1* in THLE2 cells, folate and lipid levels were measured by ELISA and staining techniques, respectively. In addition, up to 482 amino acids and lipid metabolites were semi-quantified in *SLC19A1*-knockdown (KD) cells through ultra-high-performance liquid chromatography coupled with mass spectrometry. SNPs, rs1051266 and rs3788200, were significantly associated with the development of fatty liver for the single-marker allelic test. The minor alleles of these SNPs were associated with a 0.6/−1.67-fold decreased risk of developing MAFLD. When *SLC19A1* was KD in THLE2 cells, intracellular folate content was four times lower than in wild-type cells. The lack of functional *SLC19A1* provoked significant changes in the regulation of genes associated with lipid droplet accumulation within the cell and the onset of NAFLD. Metabolomic analyses showed a highly altered profile, where most of the species that accumulated in *SLC19A1*-KD-cells belong to the chemical groups of triacylglycerols, diacylglycerols, polyunsaturated fatty acids, and long chain, highly unsaturated cholesterol esters. In conclusion, the lack of *SLC19A1* gene expression in hepatocytes affects the regulation of key genes for normal liver function, reduces intracellular folate levels, and impairs lipid metabolism, which entails lipid droplet accumulation in hepatocytes.

## 1. Introduction

During the last decade, genetic association studies have proved useful in identifying biomarkers of susceptibility to complex multifactorial diseases, including non-alcoholic fatty liver disease (NAFLD), currently termed metabolic-associated fatty liver disease (MAFLD) [1,2,3,4]. However, the vast majority of these studies lacked functional tests of the identified genes in the progression of the disease. Given these existing data gaps, the molecular mechanisms associated with the onset and progression of NAFLD need to be further explored. In this context, early, non-invasive identification of patients predisposed to developing NAFLD would be clinically helpful in establishing evidence-based approaches to support modifying the patient’s lifestyle in order to avoid the progression to severe liver disease and its serious clinical complications. This progress would significantly decrease morbidity associated with MAFLD, and provide innumerable benefits to the patients, their families, and society at large.

NAFLD has a prevalence estimated to be 5–30% worldwide [5]. Therefore, it has become one of the major causes of chronic liver disease in the world [6], primarily due to the ever-increasing incidence of obesity and Type 2 Diabetes *mellitus* (T2DM). As the frequency of NAFLD increases, non-alcoholic steatohepatitis (NASH)-related cirrhosis and HCC continue to be major healthcare concerns, as well as increasing the risk for potential liver transplantation [7]. NAFLD spans from the simple accumulation of fat in the liver (steatosis) to the more severe necroinflammatory complication referred to as NASH, which may progress to cirrhosis, and HCC, entailing increased morbidity and mortality [6].

The hypothesized mechanisms to explain how steatosis progresses to NASH and HCC include non-esterified fatty acid (NEFA)-induced endoplasmic reticulum stress and consequent cellular apoptosis, oxidative stress, altered DNA methylation, and disrupted methionine metabolism [8,9]. Nevertheless, there is still no consensus as to how this actually happens. It was demonstrated that low-dose folic acid supplementation decreases plasma homocysteine in patients with cardiovascular or neuronal alterations [10]. Xia and co-workers [11] analyzed the relationship between folate concentration and its impact on steatosis. They found low serum folate levels in the Chinese population with NAFLD. Consistent with this observation, methionine metabolism is altered in patients with liver disease [9,12], and both increases or decreases in S-adenosylmethionine (SAM), the main biological methyl donor, induce mice to spontaneously develop NAFLD [13]. Studies in humans and experimental animals suggest that perturbations in one-carbon metabolism may play a role in liver damage and hepatocarcinogenesis [14,15], although these studies reported inconsistent findings that may be influenced by sex, age, ethnicity, and body weight [16,17]. In the previously mentioned studies, it has not yet been established whether low folate levels were due to low dietary intake and/or whether they were secondary to some enzymatic deficiencies in one-carbon metabolizing enzymes. It does raise the prospect that genetically determined defects in one-carbon metabolism or transport proteins could play a significant role in the development of steatosis.

The reduced folate carrier (RFC1; *SLC19A1*) provides a low affinity transport route (*K*_m_ = 2–7 μM) for folate uptake at physiological pH by operating as an antiporter exchanging folates with intracellular organic phosphates [18,19]. Functional expression of all folate transporters has been reported to be modulated by nuclear receptors. These receptors belong to a large superfamily of DNA-binding transcriptional factors that regulate tissue expression of target genes in response to specific ligand activators [19]. Variants of this gene have been previously associated with increased risk for congenital malformations such as orofacial clefts, conotruncal heart defects, spina bifida, and congenital heart disease [20,21,22].

Considering the mentioned concerns, the aim of the present study was to understand the role of *SLC19A1* gene and RFC1 protein in the development of MAFLD. Therefore, we planned (i) to study the potential genetic association of some *SLC19A1* single nucleotide variants (SNPs) with the small cohort of MAFLD patients and unaffected controls that we have in our consortium and their clinical traits’ values; as well as (ii) to generate in vitro *SLC19A1*-knockdown (KD) hepatocytes to characterize the phenotype, transcriptomic, and metabolomic effects of this deficiency.

## 2. Experimental Procedures

### 2.1. Human Subjects

After informed consent, following the ethical guidelines of the 1975 declaration of Helsinki as revised in 1983, phenotype classification was performed under the criteria defined by Brunt et al. [23], and the scoring system followed was the one described by Kleiner et al. [24]. Clinical characteristics of the participants in this study (cases and controls) are summarized in Appendix A.

Whole blood samples from 452 NAFLD patients were provided by investigators at the collaborating hospitals. All NAFLD cases were biopsy-proven and non-insulin resistant. DNA from an additional 304 control individuals was provided by the same hospitals or purchased from the DNA bank of the BIOEF Foundation (Sondika, Spain). All control samples conformed to the following inclusion criteria: (i) absence of Insulin Resistance Syndrome (no traces of hyperglycaemia, hyperlipidaemia, hypertension, or obesity); (ii) normal liver enzyme activity, as determined by measuring the levels of transaminases (AST and ALT), gamma glutamyl transferase (gGT), total bilirubin concentration, and (iii) body mass index (BMI) ≤ 30 kg/m^2^.

DNA was isolated from whole blood samples using QIAamp DNA blood Mini Kit (Qiagen), and selected DNA samples from the biobank were isolated from filter paper bloodspots and amplified with REPLIg Mini Kit (Qiagen). DNA concentration and quality were measured by spectrophotometry and agarose gels. Twenty nanograms of the whole genome amplified DNA was used for each individual characterization.

### 2.2. Selection of Genes and SNPs

Preliminary studies performed in our laboratory (data not published) identified new genetic variants (SNPs) within genes of the one-carbon metabolic pathway that appear to be associated with NAFLD. Based upon these preliminary findings and a comprehensive literature review [25,26], we selected three SNPs (rs1051266, rs3788200, rs3788190) within *Solute carrier family 19 (folate transporter) member 1* (*SLC19A1*) gene, also known as *Human reduced folate carrier 1* (*RFC1*). For the selected polymorphisms, we evaluated any possible association with susceptibility to developing NAFLD.

### 2.3. Genotyping and Association Study

SNP genotyping was performed using predesigned TaqMan Assays (Thermo Fisher Scientific) as fully described in the Supplementary Material.

The genotypes and allele frequencies obtained from NAFLD versus control individuals were compared using Haploview [27] and PLINK [28] Software. The analysis was performed using an allelic test of single-marker and multi-marker association including all individuals. Data filtering criteria were minor allele frequency (MAF) ≥ 0.01 and Hardy-Weinberg equilibrium (HWE) ≥ 0.001. Calculation of *r*^2^ and linkage disequilibrium (LD)-block estimation were analyzed in Haploview [27] (MAF ≥ 0.01 and HWE ≥ 0.001).

Quantitative continuous clinical traits (age, BMI, and blood levels of triglycerides [TG], AST, ALT, gGT, glucose, insulin, total cholesterol, and bilirubin) medians were correlated to each of the three possible genotypes for the studied SNPs in the whole population. Kruskal–Wallis test (K–W chi-squared) was applied to infer the relationship between the obtained genotypes and the above-mentioned clinical parameters.

### 2.4. Gene Knockdown in THLE2-Cells

THLE2 human liver cell line (ATCC^®^ CRL-10149^™^) [29] was purchased from ATCC^®^. This cell line was derived from normal, healthy, primary human liver cells, and their phenotype is typical of normal adult liver epithelial cells. Therefore, THLE2 are a suitable model system for our experiments since primary hepatocytes may dedifferentiate during the silencing process.

THLE2-cells were transfected with SureSilencing shRNA targeting *SLC19A1* and a negative control shRNA (336314 KH10303P, Qiagen) as described in the Supplementary Material. Each plasmid carries a puromycin resistance gene for the selection of the knockdown (KD) cells.

After selection, KD of the targeted gene was assessed by measuring gene and protein expression levels by qPCR and Western-Blot, respectively (Appendix A).

### 2.5. Folate Concentration in THLE2-Knockdown Cells

Folate concentrations within the cultured cells were determined by ELISA as previously reported by Salojin et al. [30] and compiled in the Supplementary Material.

### 2.6. Phospholipid and Neutral Lipid Droplet Staining

THLE2 transfected cells were seeded in 96 well plates and allowed to settle overnight. The following day, cells were fixed with 3.7% formaldehyde, and stained with HCS LipidTOX^TM^ green neutral lipid stain 495/505 (Life Technologies). Images were taken with an Operetta^®^ High-Content Imaging System (PerkinElmer), and fluorescence intensity was quantified with Harmony^®^ analysis software (PerkinElmer).

### 2.7. Pathway-Focused Gene Expression Analysis. PCR Arrays in Knockdown THLE2-Cells

Total RNA was isolated from triplicates of shSLC19A1-KD and shControl THLE2-cells, with the RNeasy Micro Kit (Qiagen). After assessing yield and quality of the RNA with the 2100Bioanalyzer (Agilent Technologies), cDNA synthesis and elimination of genomic DNA was carried out with RT^2^first-strand kit (Qiagen). Subsequently, gene expression profiling was performed with the RT^2^profiler^TM^ PCR-Array System (PAHS-157Z Fatty liver, Qiagen), RT^2^SYBR-Green mastermix, and amplification was performed using the Life Technologies 7900HT Fast Real-Time-PCR System thermocycler. Data were evaluated using “RT^2^profiler^TM^ PCR-Array data analysis software” (Qiagen), which calculates relative expression using the ΔΔCt method [31]. *RPLP0* and *GAPDH* were selected as reference genes (based on the small variation in Ct-values across the included samples).

Reviewing the list of the genes that were found to be differentially expressed, considering thresholds at a fold-change of 1.4 and *p*-value < 0.05, and adding to the list the gene that was silenced, *SLC19A1*, a “Reactome” functional interaction network was built using the Cytoscape software [32] (see the Supplementary Material for further details).

### 2.8. Culture of THLE2-Cells in Folic Acid Depleted Medium

THLE2 human liver cells were plated in 24-well plates and allowed to incubate overnight in complete media. The following day, media was changed for folate-free RPMI 1640 (Gibco^®^) supplemented with 10% dialyzed-FBS (Gibco^®^) and antibiotics. Reference control cells were supplemented with 0.1 mg/mL of folic acid (Sigma) and/or 0.1 mg/mL of 5-methyltetrahydrofolate (5-MTHF). The media was changed every three days. Lipid accumulation was measured every 24 h from day 3 to day 6 by staining cells with LipidTOX^TM^ as previously described.

#### 2.8.1. Metabolite Extraction for Ultra-High Pressure Liquid Chromatography Coupled to Mass Spectrometry (UHPLC-MS) Analysis

Metabolomic analytical methods, including chromatographic separation and mass spectrometric conditions are fully described in the Supplementary Material and referenced in [33,34]. Briefly, identified ion features in the methanol extract platform included fatty acids, bile acids, steroids, and lysoglycerophospholipids. Chloroform/methanol extract platforms provided coverage over glycerolipids, cholesterol esters, sphingolipids, and glycerophospholipids.

#### 2.8.2. Metabolomics Data Processing, Normalization, and Data Analysis

Data, as defined by retention time, mass-to-charge ratio pairs (Rt-m/z) were processed using the TargetLynx application for MassLynx V4.1 (Waters Corp.). Metabolite identification and linear detection range were performed as detailed in the Supplementary Material and in [33,34]. Lipid and amino acid nomenclature follows the LIPID MAPS convention, www.lipidmaps.org (accessed on 30 April 2015) and the Human Metabolome Database (HMDB), http://www.hmdb.ca/ (accessed on 30 April 2015), respectively.

Normalization factors were calculated for each metabolite following the procedure described by Martínez-Arranz et al. [35]. Obtained integration intensities within each individual sample were divided by the result of adding up all the signals integrated in the study (all mass peaks) as described in [34]. Described procedures were developed with R Software (R V3.2.0_2010; http://cran.r-project.org (accessed on 30 April 2015)).

Principal components analysis (PCA) was performed by exporting the normalized data to SIMCA-P+ (V14.1, Umetrics). The metabolites that discriminate between *SLC19A1*-KD and control THLE2-cells in a multivariate manner are in Appendix A. Additionally, in order to find the highest differences in the concentration of each metabolite between the two groups, that is, univariate analysis, we implemented an unpaired Student’s *t*-test.

## 3. Results

### 3.1. SNPs Located within SLC19A1 Are Associated with NAFLD Susceptibility

Following TaqMan SNP genotyping in a cohort of 756 individuals (452 NAFLD patients and 304 controls) two out of the three of the targeted SNPs (rs1051266 and rs3788200) proved to be significantly associated with a NAFLD diagnosis (*p* < 10^−3^) in the single-marker allelic test (Appendix A). The certainty that the studied population did not present any stratification is shown in the Supplementary Material in our previous study [3], and confirmed by the statistical data analyses. These two *SLC19A1* SNPs showed to be in strong linkage disequilibrium (LD = 0.98) in the studied population; therefore, they appear to be non-randomly associated. Allele frequency of rs3788190 showed significant deviation from Hardy Weinberg equilibrium (HWE < 0.001) and, thus, this SNP was dropped from further consideration.

Genotypic association for these two significant SNPs was also tested. The genotype distribution of rs1051266 (AA/AG/GG) was 78/141/68 for controls and 84/192/150 for cases, whereas rs3788200 (GG/GA/AA) was 77/145/75 for cases and 79/205/156 for NAFLD patients. Both associations showed a *p*-value lower than 10^−2^ (0.002194 and 0.003351, respectively).

The association of rs1051266 and rs3788200 genotypes in NAFLD-cases and controls was tested for interactions with their clinical phenotype. Significant relationships (Kruskal–Wallis *p* < 0.05) were observed among specific genotypic variants and BMI, TG, ALT, and gGT levels, as well as with the AST/ALT ratio (Table 1).

### 3.2. SLC19A1-Knockdown THLE2-Cells Significantly Exhibit Reduced Intracellular Folate Concentrations

The cellular folate concentration in *SLC19A1*-KD THLE2 cells was found to be four-fold lower than in the control cells (THLE2 with a scramble shRNA control), with a *p*-value of 1.05 × 10^−5^ (Figure 1).

#### 3.2.1. Pronounced, Spontaneous Lipid Droplet Accumulation in SLC19A1-Knockdown THLE2-Cells

Following two weeks of selection, *SLC19A1*-KD cells showed refringent granules when observed under a phase-contrast microscope. To confirm the presence of lipid droplets, we stained THLE2-cells with LipidTOX. *SLC19A1*-KD cells showed three-times more accumulation of neutral lipids than did the controls (Figure 2).

#### 3.2.2. Lipid Droplet Accumulation in the SLC19A1-Knockdown Cells Can Be Replicated by Culturing Cells in Low Folate Medium

To elucidate whether the observed lipid droplet accumulation in the *SLC19A1*-KD cells was due to low intracellular folate concentrations, control THLE2 human liver cells were cultured with folate-free media. After four days of culture, cells presented a more significant increase in lipid droplet accumulation than those grown in the same media but supplemented with folic acid and/or 5-MTHF (Figure 3).

#### 3.2.3. Effect of SLC19A1-Knockdown in THLE2-Cells Gene Expression

The whole genome expression profile of si*SLC19A1*-THLE2-cells was characterized using RT² Profiler™ PCR Array Human Fatty Liver (Qiagen, PAHS-157Z). Of the 84 genes evaluated, 44 were shown to be altered in the silenced cells. Using +/−1.4 fold-change and *p*-value < 0.05 as cut-off values for selection, the expression of 17 genes were found to be downregulated, whereas 16 other genes were upregulated (Appendix A and Figure 4A). According to gene ontology analysis, the regulated genes are involved in cholesterol and lipid metabolism (14 genes); apoptosis (5 genes); non-insulin-dependent diabetes *mellitus* (NIDDM) (5 genes); inflammatory responses (4 genes); insulin signalling (4 genes); carbohydrate metabolism (3 genes), and adipokine signalling (2 genes).

A reactome functional interaction network was inferred taking the list of the genes that were found to be differentially expressed and adding the gene that was silenced, *SLC19A1* (Figure 4B). This analysis showed that the downregulated *SLC19A1* is directly related to the expression of the cAMP-responsive element-binding protein 1 (*CREB1*), which is one of the central node genes in the revealed reactome.

#### 3.2.4. The Lack of SLC19A1 Results in Global Alteration of the Lipid and Amino Acid Cellular Profile

Aiming to detect the precise metabolite species whose concentrations change due to folate deprivation; metabolomics analyses were performed. Subsequently, the levels of 482 lipids and amino acids were compared in *SLC19A1*-KD cells with their matched controls (Appendix A). Analysed compounds comprised 27 amino acids, 73 fatty acyls, including NEFA, fatty amides, fatty esters, and oxidized fatty acids, 7 diglycerides (DG), 69 TG, unesterified cholesterol, 14 cholesterol esters (ChoE), 5 bile acids, 3 steroid sulfates, 225 glycerophospholipids, covering phosphatidylethanolamines (PE), phosphatidylcholines (PC), phosphatidylinositols, lyso-PE, lyso-PC, lyso-PI, and lyso phosphatidylglycerol), and 60 sphingolipids (ceramides, sphingomyelins, and monohexosylceramides).

The selected metabolites had a coefficient of variation lower than 0.15 after strict analysis filtering of metabolomics considering the small number of cell culture replicates. We found an elevated number of metabolites (356 out of 482) with different concentrations between *SLC19A1*-KD cells and controls.

The heatmap in Figure 5 represents the fold-changes and *p*-values associated with the comparison of *SLC19A1*-KD hepatocytes and shControl-THLE2. Individual and metabolite chemical group data statistical analysis is shown in Table 2. The majority of the species that accumulated in *SLC19A1*-KD cells belong to the chemical groups of TG, DG, PUFA, and long chain, highly unsaturated ChoE. More precisely, the concentration of TG(52:4); TG(16:0 + 20:4 + 16:0) and the combination of TG(58:9); TG(20:4 + 20:4 + 18:1) and TG(22:6 + 18:1 + 18:2) increased above 25-fold in *SLC19A1*-KD hepatocytes (Appendix A, box-plots). Moreover, within the TG whose concentration increased the most, those with fatty acids 20:4 and 22:5 were the most frequent.

In contrast, monohexosylceramide CMH(d44:1); CMH(d18:1/24:1), and oxidized glutathione (GSSG) decreased more than 90%. In addition, the concentration of the vast majority of amino acids in the *SLC19A1*-KD hepatocytes decreased when compared to controls, and the level of methionine, directly implicated in the folate pathway, decreased 62%.

The lowest *p*-values (below *p* < 10^−8^) were found for PC, such as PC(16:1e/20:4) which increased 3.2-fold and PC(38:5), the combination of PC(18:0/20:5) and PC(16:0/22:5), which decreased 44%, and to a lesser extent, for PE. Interestingly, most of the PC and PE whose levels increased contained the fatty acid 20:4 or 22:4 in the C-2 position.

The multivariate PCA model (Appendix A) displays a very clear separation of *SLC19A1*-KD cells and their controls, as the first principal component explained 80% of the variation of the data. This PCA shows goodness of fit scores plot (A = 1), R^2^X = 0.791, R^2^Y = 0.695. Loadings plot (Appendix A) displays the metabolites that differentiate between control THLE2-cells and *SLC19A1*-KD hepatocytes. The majority of the species that increase in *SLC19A1*-KD hepatocytes belong to the chemical groups of TG, primary fatty amides (FAA), and oxidized fatty acids (OxFA). In contrast, a large amount of LPE and amino acids decrease in RFC1 silenced cells when compared to controls. The metabolites that best differentiate between the two groups in the multivariate analysis were the glycerophospholipids PE (16:1e/20:4), PC (22:4/20:4), PC (17:0/20:3), and PC (18:1e/20:4).

## 4. Discussion

This study demonstrated the association among two sequence variants of the *SLC19A1* gene, the onset of NAFLD in human patients, and the potential molecular mechanisms underlying its development. Since the reduced folate carrier 1 (RFC1), a membrane protein encoded by *SLC19A1*, is a transporter involved in regulating the concentration of folate, it is reasonable to expect a shift in several cellular processes when its expression is altered [36]. Low folate status may be a consequence of suboptimal intake, transport, or cellular utilization of folate. When considering the studied population, rs1051266 and rs3788200 allelic and genotyping associations with NAFLD were obtained.

### 4.1. Polymorphisms in Folate Transporter Genes

Polymorphisms in folate transporter genes were implicated as risk factors for certain types of birth defects, and recent evidence from human epidemiological studies demonstrated an association between the *SLC19A1*-A80G polymorphism (rs1051266) and an increased risk for neural tube [37,38], conotruncal heart defects [39,40], and other congenital heart disease [22]. DeMarco and colleagues [37] reported that, while the *SLC19A1*-A80G variant is common in the Italian population (0.47), the allelic frequency was higher among neural tube defects (NTDs) cases and their parents than in unaffected controls. Heterozygous patients and mothers have ORs of 1.72 (95% CI 0.96–3.11) and 1.86 (95% CI 0.68–5.27), respectively. Further functional studies of this missense mutation will be possible thanks to the recent description of the cell membrane folate transporter RFC1 protein structure [41], and its clinical associations in cases vs. controls will be validated in vitro in future studies.

### 4.2. Phenotyping of MAFLD Patients

In our efforts to characterize the phenotypes of MALFD patients, a quantitative trait approach revealed a significant genetic association between rs1051266 and rs3788200, and a number of clinical parameters that helped to define our study cohorts. Remarkably, a very significant association of both *SLC19A1* SNPs genotypes was found with the AST/ALT ratio (*p* = 1.04 × 10^−5^ and 9.97 × 10^−6^, respectively), which is a known biomarker of hepatic damage. Moreover, these results coupled the rs1051266-GG genotype, and its linked rs3788200-AA haplotype, to be correlated with the highest BMI (µ = 40.35 kg/m^2^), the lowest blood TG (µ = 1.69 mmol/L), the highest AST and gGT (µ = 22 IU/L and 24 IU/L, respectively) and the lowest AST/ALT ratio (µ = 0.91). This classical pattern is particularly useful in differentiating between alcoholic liver disease and NAFLD, with the latter normally associated with a decreased AST/ALT ratio and elevated gGT [42]. Decreased concentration of TG in serum is not frequent in NAFLD cases, although this could imply a diminished TG secretion, with the concomitant lipid droplet accumulation in the liver, together with increased TG lipase activity in the studied patients. Certainly, these patients were histologically diagnosed as having fatty liver disease.

### 4.3. Low Folate Levels Contribute to Lipid Droplet Accumulation

The general effects of altered dietary folate levels are well known. Both low- and high-levels of folic acid similarly impact global DNA methylation, cytome biomarkers, DNA damage induced by oxidative stress, and DNA base excision repair gene expression [43]. On the one hand, low folate diets lead to steatosis in the liver of mice, secondary to altering a number of enzymes controlling the methylation cycle [25], promotes lipid accumulation and leptin production of adipocytes [44], and even provokes subsequent cognitive dysfunction in mice [45]. On the other hand, supplementation with folic acid decreases plasma homocysteine [10], attenuates hepatic inflammation in mice fed high fat diets [46], and reduces fibrosis in NASH by autophagy control [47]. There is increasing evidence that specific alterations in gut microbiota and short chain fatty acid production is correlated with dietary folic acid abdominal fat deposition reduction in broiler chickens [48].

Consistent with these observations, the suppression of the *SLC19A1* gene promoted the spontaneous accumulation of several lipid droplets as well as a decrease in the folate content in *SLC19A1*-KD THLE2-cells when compared to controls. Specifically, excess folic acid increases lipid storage, weight gain, and adipose tissue inflammation in rats fed a high fat diet [49]. Additionally, studies found a link between high folate status in mothers and insulin resistance and adiposity in children [50].

The relationship among altered folate status, hepatic lipid metabolism, and the development of NAFLD was extensively studied [25,51,52]. However, the metabolic pathways connecting this triad have only recently begun to be interrogated. Recent advances in metabolomics techniques and machine learning shed light on this subject [13,53,54], enabling us to better understand the complex crossroad of metabolism during the onset of the liver pathology.

### 4.4. Omics Analyses Unravelled a Potential Role of the Gene SLC19A1 in NAFLD

In this project, transcriptomics and metabolomics analyses were performed, revealing that the silencing of *SLC19A1* in hepatocytes regulates several gene pathways controlling lipid and amino acid concentrations. Its downregulation provoked a profound impact on the metabolome of human hepatocytes, and the storage of specific TG, ChoE, and PUFA in hepatocytes is a critical feature of NAFLD [33]. The accumulation of TG in the hepatocyte lipid droplets may be caused by the increased delivery, augmented synthesis, reduced β-oxidation of NEFA, and/or decreased TG and ChoE export through very low-density lipoproteins (VLDL) [55], and TG species were described and patented as NAFLD biomarkers [13].

The expression of several genes found in *SLC19AS1*-KD hepatocytes may help to explain some of the metabolomics results obtained. The increased gene expression of the lipogenic key enzyme fatty acid synthase (*FAS*), and the reduced X-box binding protein 1 (*XBP1*), stimulate the increase of the concentration of PUFA and cholesterol through endoplasmic reticulum stress-activated unfolding protein response, respectively [56]. The increased PUFA was also reported to be stimulated by the patatin-like phospholipase domain-containing protein 3/adiponutrin (PNPLA3), as well as the elevated concentrations of TG (especially the species 50:1) [57]. Also, the analysis of the reactome of *SLC19A1* revealed its direct relation to the core transcription factor *CREB1*, which has been proposed as a potential therapeutic target for liver disease [58].

Together, these changes would promote the development of NAFLD through the modulation of the metabolic pathways involved in TG metabolism. As previously observed in both human and murine models, species analysis of the TG precursor DG demonstrated multiple fold increases in relatively short chain species [59].

Some SNPs in *SLC19A1* are known to be associated with cholesterol level control [60]. In this case, altered expression of genes controlling lipoprotein metabolism, such as *ABCA1*, *APOA*, and *APOE*, may interfere with the accumulation of ChoE within *SLC19A1*-KD lipid droplets. Furthermore, the observed decreased methionine and GSSG levels could be a direct consequence of the low folate present in *SLC19A1*-KD cells. These compounds would indicate reduced cellular methylation status that affect thiol disulphide exchange, cellular signalling, and detoxification. Therefore, changes in GSH levels and oxidative stress can impair multiple hepatic functions [61].

The expression of the gene *IL-6* was found to increase by 41% in the *SLC19A1*-KD cells ( Appendix A, FC = 1.41 and *p* = 0.048). This interleukin has been markedly increased in the livers of patients with NASH as compared to patients with simple steatosis. Furthermore, it was suggested to play an important role in NASH development as well as in systemic insulin resistance and diabetes [62]. Interestingly, the expression of IL-6 has been also elevated in severe COVID-19 patients [63].

The most significant changes detected in the composition of *SLC19A1*-KD cells were found for an elevated number of PC. In the liver, PC can be formed either through the Kennedy pathway or directly from PE by the enzyme PE N-methyltransferase (PEMT) [64]. PC formed via the PEMT pathway are primarily enriched in long-chain PUFA, such as docosahexaenoic acid (22:6n-3) [65,66]. Accordingly, a decrease in the ratio PC(22:6n-3) to total PC was identified in *SLC19A1*-KD cells of mice, suggesting a decreased synthesis of PC. Consistent with this result, *Mthfr*^+/−^ mice were reported to present with reduced PEMT expression in their livers [67].

Decreased PEMT-derived PC synthesis can contribute significantly to hepatic TG deposits due to impaired secretion of abnormal VLDL lipid particles [66]. Mass spectrometry analyses found that an elevated number of TG were hoarded within hepatocyte lipid droplets, and that the levels of concrete TG species augmented in a very marked manner due to *SLCA19A1* gene silencing. These findings suggest that folate deficiency, secondary to decreased expression of *SLCA19A1* gene, would moderate de novo PC synthesis, resulting in the storage of hepatic TG. Furthermore, the metabolites and genes found to be highly altered in hepatocytes with folate and lipid droplet accumulation represent a high value target for future research aiming to unravel the molecular physiopathology of MAFLD.

These new data should be useful in the search for biomarkers of MAFLD, which requires a liver biopsy to be confirmed. The research reported could help to decipher some new molecular pathways involved in the development of MAFLD, opening new horizons for drug development and disease diagnosis.

## 5. Conclusions

Herein, we report the first evidence of genetic variants of *SLC19A1* associated with the onset of MAFLD in humans. The altered equilibrium of folate homeostasis in THLE2-hepatocytes dysregulates other genes critical for normal liver function, causing low intracellular folate levels and spontaneous lipid droplet accumulation.

## Figures and Tables

**Figure 1 biomedicines-11-00337-f001:**
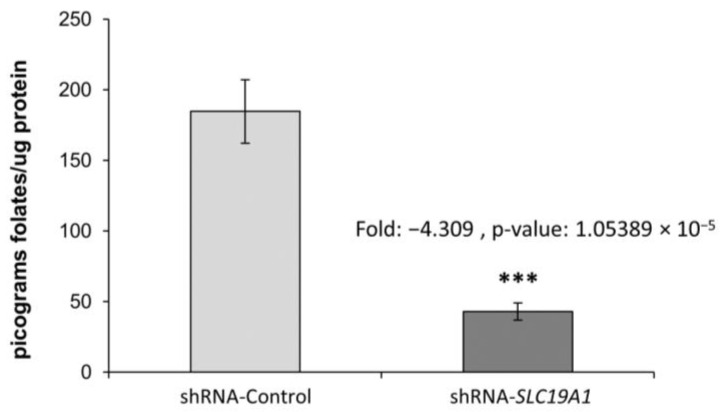
Cellular folate content in *SLC19A1*-KD cells and in their unaffected controls. Fold-change: −4.309, *p*-value: 1.05389 × 10^−5^. ***: Difference between shRNA-Control and shRNA-*SLC19A1* is statistically significant, p-value ≤ 0.001.

**Figure 2 biomedicines-11-00337-f002:**
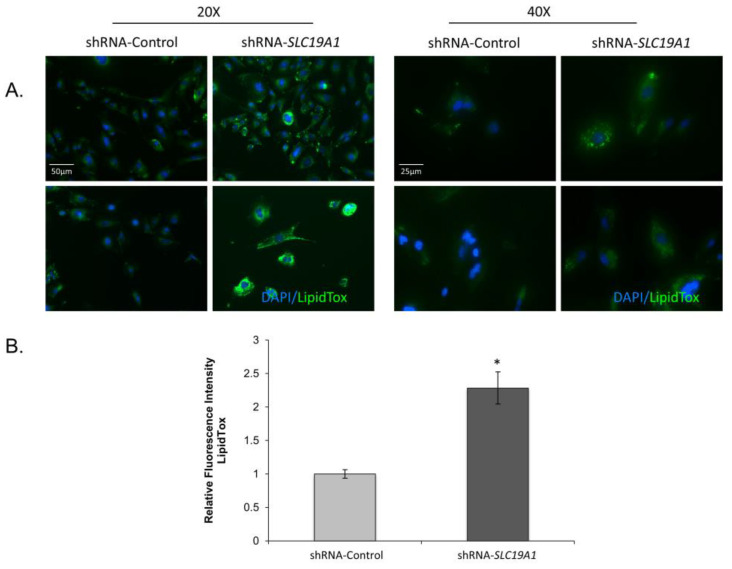
Lipid droplet accumulation in *SLC19A1*-knockdown THLE2-cells. *SLC19A1*-KD THLE2-cells develop steatosis spontaneously and show impaired lipid metabolism. DAPI (blue staining) for nucleic acids and LipoTox (green staining) for lipids. Scale bar added to 20× and 400× images. *: Difference between shRNA-Control and shRNA-*SLC19A1* is statistically significant, p value<0.05.

**Figure 3 biomedicines-11-00337-f003:**
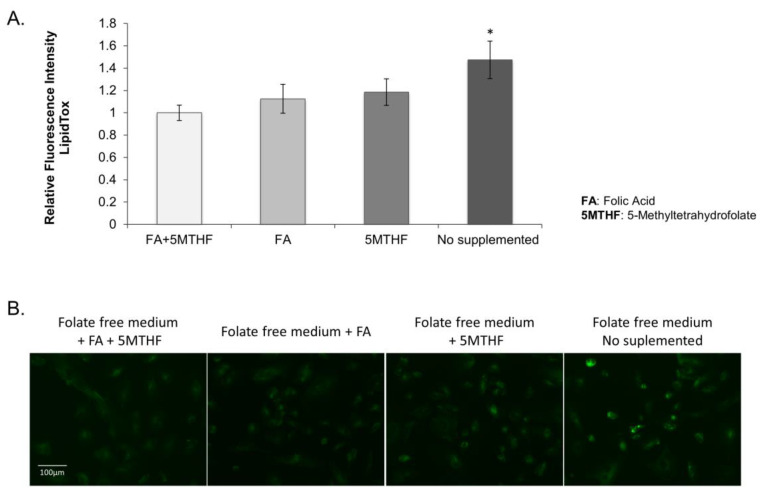
THLE2 human liver cells cultured on folate-free RPMI supplemented with 10% dialyzed FBS. (A) Graphical representation of the fluorescence mean calculated in three wells per each treatment and three independent fields per well. *: Difference between fluorescence on “no supplemented cells” vs “cells supplemented with FA and/or 5MTHF” is statistically significant, *p* value < 0.05. (**B**) Images of the different culture conditions stained with LipidTOX^TM^ (green) and DAPI (blue). Scale bar = 100 uM (10× objective).

**Figure 4 biomedicines-11-00337-f004:**
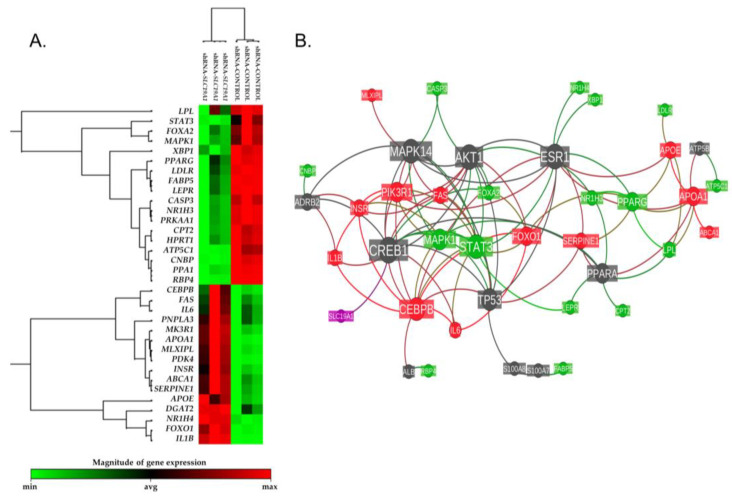
Differential gene expression heatmap and reactome functional interaction network. (**A**) Heatmap representation of the expression levels detected for the 44 genes shown to be altered in the silenced cells (+/−1.4 fold-change and *p*-value < 0.05 as cut-off values for selection). (**B**) Reactome functional interaction network of the same 44 genes. The size of each gene’s node represents the number of connections with the other studied genes within the reactome’s network. Nodes’ color is coded as follows: in red, up-regulated genes; in green, down-regulated genes; in dark grey, linker genes; in magenta, the *SLC19A1* gene. Each bond line is colored based on the regulation of the two nodes it connects.

**Figure 5 biomedicines-11-00337-f005:**
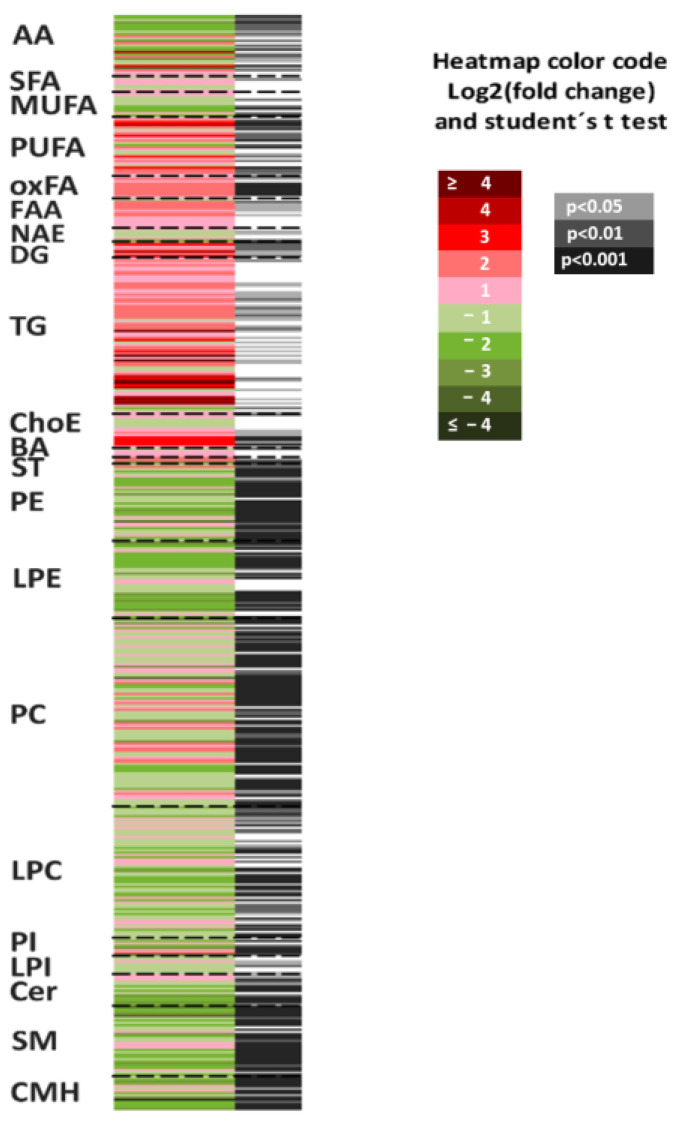
Metabolomics fingerprint of *SLC19A1*-KD cells. Heatmap represents metabolomic signatures of the comparison of *SLC19A1*-KD and shControl THLE2-cells. Metabolites present in the picture were ordered according to their carbon number and unsaturation degree. In the second column, grey lines correspond to significant fold-changes of individual metabolite levels, darker grey color stress higher significances (Student’s *t*-test *p* < 0.05, *p* < 0.01, or *p* < 0.001).

**Table 1 biomedicines-11-00337-t001:** Correlation of the rs1051266 and rs3788200 genotypes vs. clinical quantitative traits values for the whole studied population (cases plus controls).

Clinical Traits	rs1051266 Genotype	Median	rs1051266 *p*-Value	rs3788200 Genotype	Median	rs3788200 *p*-Value
**BMI (kg/m^2^)**	AA	29.37	**0.0021**	GG	28.1	**0.0009**
	AG	34.65		GA	35.75	
	GG	40.35		AA	40.35	
**Triglycerides (mmol/L)**	AA	1.89	**0.0128**	GG	1.9	**0.0164**
	AG	1.89		GA	1.9	
	GG	1.69		AA	1.69	
**AST/GOT (IU/L)**	AA	22	0.4921	GG	22	0.2502
	AG	21		GA	21	
	GG	20.5		AA	20	
**ALT/GPT (IU/L)**	AA	19	**0.0142**	GG	19	**0.0498**
	AG	21		GA	21	
	GG	22		AA	22	
**AST/ALT ratio**	AA	1.14	**1.04 × 10^−5^**	GG	1.12	**9.97 × 10^−6^**
	AG	0.97		GA	0.97	
	GG	0.91		AA	0.91	
**gGT (IU/L)**	AA	19.5	**0.0258**	GG	20	**0.0298**
	AG	21		GA	21	
	GG	24		AA	24	
**Glucose (mmol/L)**	AA	5.72	0.4609	GG	5.72	0.5239
	AG	5.5		GA	5.5	
	GG	5.56		AA	5.55	
**Insulin (mIU/L)**	AA	13.3	0.6585	GG	13.95	0.7243
	AG	13.15		GA	13.5	
	GG	13.55		AA	13.5	
**Total Cholesterol (mmol/L)**	AA	4.71	0.3726	GG	4.73	0.7426
	AG	4.84		GA	4.83	
	GG	4.72		AA	4.76	
**Bilirubin (µmol/L)**	AA	8	0.567	GG	8.21	0.3116
	AG	7.01		GA	7.01	
	GG	7.52		AA	7.35	
**Age (years)**	AA	36	0.5703	GG	36.5	0.7286
	AG	36		GA	36	
	GG	37		AA	37	

Median values of each clinical trait are detailed for each SNP genotype. Kruskal–Wallis test (K–W chi-squared) was applied to infer the relationship between the obtained genotypes and clinical values (non-parametric). Significant correlations (*p*-value < 0.05) are marked in **bold**.

**Table 2 biomedicines-11-00337-t002:** Metabolomic analysis of *RFC1* silenced cells compared with their controls. Significant differences are represented in bold letters.

Chemical Group	Class	Individual Notation	Student’s *t*-Test (*p*)	Log_2_ (Fold Change)	Fold Change
AA	Amino acids	Amino acids	**1.10 × 10^−4^**	−1.61	0.33
NEFA	Non-esterified fatty acids	Non-esterified fatty acids	**2.64 × 10^−3^**	0.85	1.80
SFA	Non-esterified fatty acids	Saturated fatty acids	8.82 × 10^−1^	0.08	1.06
MUFA	Non-esterified fatty acids	Monounsaturated fatty acids	1.55 × 10^−1^	−0.16	0.89
PUFA	Non-esterified fatty acids	Polyunsaturated fatty acids	**1.06 × 10^−2^**	1.32	2.49
oxFA	Oxidized fatty acids	Oxidized fatty acids	**1.87 × 10^−4^**	1.45	2.73
FAA	Fatty amides	Primary fatty amides	5.28 × 10^−2^	0.99	1.98
NAE	Fatty amides	N-Acyl ethanolamines	**3.25 × 10^−2^**	−0.39	0.76
DG	Glycerolipids	Diglycerides	**3.35 × 10^−3^**	1.35	2.56
TG	Glycerolipids	Triglycerides	**3.49 × 10^−2^**	0.73	1.65
ChoE	Sterols	Cholesterol Esters	**1.30 × 10^−2^**	1.43	2.69
BA	Bile Acids	Bile Acids	4.75 × 10^−1^	−0.18	0.88
ST	Sterols	Steroids	**2.12 × 10^−3^**	−0.84	0.56
PE	Glycerophospholipids	Phosphatidylethanolamines	**1.93 × 10^−5^**	−0.24	0.85
LPE	Glycerophospholipids	Lysophosphatidylethanolamines	**5.14 × 10^−5^**	−1.19	0.44
PC	Glycerophospholipids	Phosphatidylcholines	**5.03 × 10^−3^**	−0.05	0.97
LPC	Glycerophospholipids	Lysophosphatidylcholines	**8.39 × 10^−3^**	−0.94	0.52
PI	Glycerophospholipids	Phosphatidylinositol	**1.33 × 10^−2^**	0.34	1.27
LPI	Glycerophospholipids	Lysophosphatidylinositol	6.80 × 10^−2^	−0.28	0.83
LPG	Glycerophospholipids	Lysophosphatidylglycerol	**1.10 × 10^−2^**	−0.96	0.52
Cer	Sphingolipids	Ceramides	**3.35 × 10^−5^**	−1.06	0.48
SM	Sphingolipids	Sphingomyelins	**4.81 × 10^−7^**	−0.67	0.63
CMH	Sphingolipids	Monohexosylceramides	**2.04 × 10^−6^**	−1.63	0.32

Significant correlations (*p*-value < 0.05) are marked in **bold**.

## Data Availability

Obtained data are all included in tables or supplementary tables of this submission. Potential intermediate data described and discussed in the manuscript, but not shown in tables or figures, will be made available upon request.

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
