# Peer review of "Impaired Function of Solute Carrier Family 19 Leads to Low Folate Levels and Lipid Droplet Accumulation in Hepatocytes"

_biomedicines, 2023, doi:10.3390/biomedicines11020337_

Round 1
Reviewer 1 Report
I have read with interest the paper received, entitled “Impaired Function of Solute Carrier Family 19 Leads To Low Folate Levels And Lipid Droplet Accumulation In Hepatocytes”. In this work, Cano and co-workers are interested in analysing the connections between NAFL and folate levels and focus their research on SLC19A1, a transporter of folate involved in the regulation of intracellular concentrations of folate. Authors downregulate this gene and are able to prove through metabolomic and transcriptomic analysis a correlation between lower levels of folate and a higher accumulation of steatosis in an hepatocyte cell line. Authors also perform genetic association studies to test the association of three SNPs within the SLC19A1 gene and find that two genetic modifications significantly associated to the presence of NAFL and BMI.
General concept comments:
1. I recommend authors to try to simplify table S1 (Two columns containing mean ± SD for each group) and include statistical tests to determine which clinical variables differ between populations under study.
2. It would be of extreme interest to check if the genetic association is associated not only to the disease itself, but to relevant features of disease stage and progression. As all patients are biopsy proven, I was wondering if it would it be possible to perform stratified analysis of SNP association against relevant features of the disease (presence of steatohepatitis, NAS score, fibrosis stage).
3. I recommend authors to consider fusing the data presented in tables 1 and S2 (pages 8 and 9) in a new table in which similar analysis are applied to all clinical variables. Multivariate testing analysis including all variables significantly associated in the univariate analysis is also highly recommended, as it will help confirming their independent association to the genetic markers.
4. I would strongly recommend authors to consider performing some kind of multiple testing correction (FDR, Bonferroni, etc.) in the transcriptomics and metabolomic analysis.
5. I would strongly recommend authors to consider simplifying their tables and figures (there are 45 pages of a table which should be supplementary data within the results description), clustering them where appropriate (e.g. cell lines characterisation [Cluster figures S1, 1-2-3] can be a single figure) and re-considering the priority given to some of them (e.g. SNP results for NAFL are included in table S2, which I assume it´s supplementary data, while the rest of clinical variables are included in table 1).
6. I have noted some references missing. Please consider including them and discussing them where appropriate (DOI:https://doi.org/10.1016/j.jhep.2022.06.033, ttps://doi.org/10.1371/journal.pone.0202910, https://doi.org/10.1016/j.aninu.2022.08.013, https://doi.org/10.3390/nu8100594, https://www.ncbi.nlm.nih.gov/pmc/articles/PMC9083361/)
7. I would advise authors to consider rephrasing the following sentences:
a. “Low serum folate levels are a risk factor for developing metabolic associated fatty liver disease (MAFLD)”. … “They found that low serum folate levels were a risk factor for NAFLD in the Chinese population”… “They found that low serum folate levels were a risk factor for NAFLD in the Chinese population”… I am not really sure about if there is evidence enough to claim that low folate levels are a risk factor for NAFL. Correlation does not imply causation. Have low folate levels proven to truly be a risk factor for disease development in studies including several evaluations throughout time? If not, and even if there is an inverse association (lower levels in NAFL patients), which by the way has been recently proven in a US population too (doi: 10.3390/nu14061224”), I would recommend authors to replace this sentence by a more appropriate one depicting what has been demonstrated to date.
b. “It has been demonstrated that low-dose folic acid supplementation decreases plasma homocysteine in patients with cardiovascular or neuronal alterations, preventing those diseases” Reference provided does not prove a direct association between folic acid supplementation and the prevention of diseases. I would consider rephrasing or including additional references able to strengthen and/or support this sentence.
c. “Recently, MAFLD has been found to be a potential risk factor for SARS-CoV-2 infection and severe COVID-19, independent of metabolic syndrome[18], along with lower folate levels, being consistently associated with increasing stages of COVID-19 severity[19]. Covid-19 is not addressed in any way in the research described in the paper. I would recommend authors to consider eliminating this section.
d. “These new data would also raise the definition of a set of biomarkers for non-invasive diagnosis to replace the current reference status that requires a liver biopsy to evaluate most cases of liver disease. Please consider avoiding overstatement.
Specific comments:
1. I think that affiliations contain text (usually in brackets) that shouldn´t be there. Probably from an earlier version. Please revise.
Author Response
Response to Reviewer 1:
We really appreciate your comments which we have used to vastly improve the revised manuscript. We have detailed below the steps we have taken to modify the manuscript or provided our justification when we had issues with the reviewer’s suggestions:
General concept comments:
- I recommend authors to try to simplify table S1 (Two columns containing mean ± SD for each group) and include statistical tests to determine which clinical variables differ between populations under study.
We have revised the Table S1, by including it within Table 1, as suggested. Thus, the Supplementary documents will be renumbered, and the former S1 Table removed.
- It would be of extreme interest to check if the genetic association is associated not only to the disease itself, but to relevant features of disease stage and progression. As all patients are biopsy proven, I was wondering if it would it be possible to perform stratified analysis of SNP association against relevant features of the disease (presence of steatohepatitis, NAS score, fibrosis stage).
The reviewer has raised a very interesting approach, one that we considered during our investigation. However, the limited statistical power that we had given the nature of our clinical sample population rendered the results non-significant. Consequently, we changed the strategy and considered the analysis of quantitative clinical traits related to MAFLD status.
- I recommend authors to consider fusing the data presented in tables 1 and S2 (pages 8 and 9) in a new table in which similar analysis are applied to all clinical variables. Multivariate testing analysis including all variables significantly associated in the univariate analysis is also highly recommended, as it will help confirming their independent association to the genetic markers.
As described above in comment 1, we have merged Table 1 and Table S1 for a clearer description of the obtained results.
Concerning the multivariate testing analysis, we were aware that several of the analyzed clinical variables are physiologically related, but multivariate testing (rank MANOVA test) did not yield any significant results (see results in the screenshot below). Thus, we proceed to the univariate testing, which, as we showed, yields very interesting significant correlations (see amended Table 1).
| Call: | |||||||
| cbind(age, | bmi, | ast.got, | alt.gpt, | glucose, | insuline, | ggt, | cholesterol, |
| triglycerids, | glycated.hemoglobin, | bilirubin, | hdl.cholesterol, | ||||
| aa.ratio) | ~ | rs1051266 | |||||
| Test: | |||||||
| Test | statistic | p-value | |||||
| rs1051266 | 10.205 | 0.162 | |||||
| Call: | |||||||
| cbind(age, | bmi, | ast.got, | alt.gpt, | glucose, | insuline, | ggt, | cholesterol, |
| triglycerids, | glycated.hemoglobin, | bilirubin, | hdl.cholesterol, | ||||
| aa.ratio) | ~ | rs3788200 | |||||
| Test: | |||||||
| Test | statistic | p-value | |||||
| rs3788200 | 12.297 | 0.095 | |||||
- I would strongly recommend authors to consider performing some kind of multiple testing correction (FDR, Bonferroni, etc.) in the transcriptomics and metabolomic analysis.
We have included FDR values in Tables S3 and S4, where multiple tests were performed.
- I would strongly recommend authors to consider simplifying their tables and figures (there are 45 pages of a table which should be supplementary data within the results description), clustering them where appropriate (e.g. cell lines characterisation [Cluster figures S1, 1-2-3] can be a single figure) and re-considering the priority given to some of them (e.g. SNP results for NAFL are included in table S2, which I assume it´s supplementary data, while the rest of clinical variables are included in table 1).
It is important that we are very clear concerning this issue, since we believe that there is some misunderstanding by the reviewer, so we have attempted to clarify our position. In the originally submitted manuscript, all tables and figures are included (manuscript and supplementary) as requested by the journal for new submissions. Thus, this is the reason why the document was so lengthy. We think that this should be taken into consideration, as we organized the manuscript according to the guidelines of the Biomedicines journal.
- I have noted some references missing. Please consider including them and discussing them where appropriate (DOI:https://doi.org/10.1016/j.jhep.2022.06.033, https://doi.org/10.1371/journal.pone.0202910, https://doi.org/10.1016/j.aninu.2022.08.013, https://doi.org/10.3390/nu8100594, https://www.ncbi.nlm.nih.gov/pmc/articles/PMC9083361/)
We appreciate these recommendations. All of the suggested references are relevant, and we have included them in the Discussion in the revised manuscript on page 52.
- I would advise authors to consider rephrasing the following sentences. a.“Low serum folate levels are a risk factor for developing metabolic associated fatty liver disease (MAFLD)”. … “They found that low serum folate levels were a risk factor for NAFLD in the Chinese population”… “They found that low serum folate levels were a risk factor for NAFLD in the Chinese population”… I am not really sure about if there is evidence enough to claim that low folate levels are a risk factor for NAFL. Correlation does not imply causation. Have low folate levels proven to truly be a risk factor for disease development in studies including several evaluations throughout time? If not, and even if there is an inverse association (lower levels in NAFL patients), which by the way has been recently proven in a US population too (doi: 10.3390/nu14061224”), I would recommend authors to replace this sentence by a more appropriate one depicting what has been demonstrated to date.
Since, as the reviewer states, correlation does not imply causation we have modified the text of the revised manuscript so that it now reads: “low serum folate levels are inversely associated with metabolic associated fatty liver disease (MAFLD)” (page 1) and “They found low serum folate levels in the Chinese population with NAFLD” (page 3).
Compared to study participants without NAFLD, those participants with NAFLD had a higher prevalence of metabolic disorders and lower dietary folate, decreased serum vitamin B12 and folate levels, and an increase in RBC folate level.
b.“It has been demonstrated that low-dose folic acid supplementation decreases plasma homocysteine in patients with cardiovascular or neuronal alterations, preventing those diseases” Reference provided does not prove a direct association between folic acid supplementation and the prevention of diseases. I would consider rephrasing or including additional references able to strengthen and/or support this sentence.
Thank you for the suggestion, and in the revised manuscript we have eliminated the affirmation “preventing those diseases” (page 3) because this evidence has not been demonstrated.
c.“Recently, MAFLD has been found to be a potential risk factor for SARS-CoV-2 infection and severe COVID-19, independent of metabolic syndrome[18], along with lower folate levels, being consistently associated with increasing stages of COVID-19 severity[19]. Covid-19 is not addressed in any way in the research described in the paper. I would recommend authors to consider eliminating this section.
This section has been eliminated, as suggested by the reviewer (see page 3).
d.“These new data would also raise the definition of a set of biomarkers for non-invasive diagnosis to replace the current reference status that requires a liver biopsy to evaluate most cases of liver disease. Please consider avoiding overstatement.
These new data would be useful in the search of biomarkers for MAFLD, which requires a liver biopsy to be confirmed and therefore is beyond the scope of our study. The reported research also helps to decipher the molecular pathways involved in the development of MAFLD and open new horizons for drug development and disease diagnosis. See amended text on page 53 of the reviewed manuscript.
Specific comments:
- I think that affiliations contain text (usually in brackets) that shouldn´t be there. Probably from an earlier version. Please revise.
We believe that affiliation details are written as required by the journal but, of course, if we are mistaken, we will accomplish all the Editor’s corrections.

Reviewer 2 Report
In this original article, the authors applied genomic, transcriptomic, and metabolomic techniques to clarified the role of SLC19A1 in MAFLD. The authors concluded that lack of SLC19A1 gene expression in hepatocytes affects the regulation of key genes for normal liver function, reduces intracellular folate levels, and impairs lipid metabolism, which leads lipid droplet accumulation in hepatocytes. This article used comprehensive strategy to reveal how SLC19A1 regulated lipid metabolism in liver and involved in MAFLD. The overall research content is of good quality. However, the formats or presentation of the results (text and tables) need to be improved.
1. Is there any difference in definition of MAFLD and NAFLD? Or just simply change the term? If the definition was different, it should be explained in the text. If their definition was the same, it would be better to synchronize with one term in the same article.
2. Supplementary tables and figures should not be inserted in the main text. If these data were important, they should be rearranged to concise formats and inserted into result section. Supplementary tables and figures should just be provided as additional files.
3. The clinical parameters (raw data) in Table S1 should be described as mean ± sd or sem, put the data of controls and cases next to each other, and explain whether there are significances in their comparison. These clinical parameters should be rearranged as Table 1 and inserted in main text and leave Table S1 as a supplementary file.
4. Table 1 in page 8, the first item “Kruskal-Wallis” should be “Clinical traits”.
5. The detail methods and results should not be explained in table footnotes and figure legends.
Author Response
Comments to Reviewer 2:
We really appreciate your comments which we have used to vastly improve the revised manuscript. We have detailed below the steps we have taken to modify the manuscript or provided our justification when we had issues with the reviewer’s suggestions:
- Is there any difference in definition of MAFLD and NAFLD? Or just simply change the term? If the definition was different, it should be explained in the text. If their definition was the same, it would be better to synchronize with one term in the same article.
There is no difference in the meaning of MAFLD vs NAFLD, and so, both terms have been used as synonyms in recent years. Experts in the field suggested using the term MAFLD for metabolic (dysfunction) associated fatty liver disease. The new definition puts increased emphasis on the important role of metabolic dysfunction. Thus, within the revised manuscript, we refer to this syndrome as MAFLD, except in the cases we mention the work of others and the corresponding articles only mention NAFLD. We had used this criterium in order to respect what the referred articles say precisely. We hope this explanation clarifies our use of these two acronyms.
- Supplementary tables and figures should not be inserted in the main text. If these data were important, they should be rearranged to concise formats and inserted into result section. Supplementary tables and figures should just be provided as additional files.
It is important that we are very clear concerning this issue, since we believe that there is some misunderstanding by the reviewer, so we have attempted to clarify our position. In the originally submitted manuscript, all tables and figures are included (manuscript and supplementary) as requested by the journal for new submissions. Thus, this is the reason why the document was so lengthy. We think that this should be taken into consideration, as we organized the manuscript according to the guidelines of the Biomedicines journal.
- The clinical parameters (raw data) in Table S1 should be described as mean ± sd or sem, put the data of controls and cases next to each other, and explain whether there are significances in their comparison. These clinical parameters should be rearranged as Table 1 and inserted in main text and leave Table S1 as a supplementary file.
We have amended the Table S1, by merging it with Table 1 following your recommendation.
- Table 1 in page 8, the first item “Kruskal-Wallis” should be “Clinical traits”.
We appreciate the reviewer point out this issue which we have corrected in the revised manuscript as seen in the new Table 1.
- The detail methods and results should not be explained in table footnotes and figure legends.
We have deleted all the methods and results descriptions from the manuscript table footnotes and figure legends, but not for the ones in the supplementary tables, since we consider important that those can be self-explanatory documents.
Please, find these changes in the amended legends for Table 2 and Figures 1, 2, 3, 4 and 5.

Reviewer 3 Report
The presented paper represents an outsstanding work and contribute well to the field. It is well written and comprehensive in its content.
There are only few questions left:
1. What is the functional effect of the analysed SNP on the protein level/function in vivo. Since in the expermental part of the paper the data are made by dramatically downregulating the SLC19A1 gen expression it would be helpful to know how the SNPs influences the gene expression.
2. Is exessive folate suplementation in the media able to reduce the effects of SCL19A1 dowregulation. This would be of clinical interest.
Author Response
Response to Reviewer 3:
We really appreciate your comments which we have used to vastly improve the revised manuscript. We have detailed below the steps we have taken to modify the manuscript or provided our justification when we had issues with the reviewer’s suggestions:
- What is the functional effect of the analysed SNP on the protein level/function in vivo. Since in the expermental part of the paper the data are made by dramatically downregulating the SLC19A1 gen expression it would be helpful to know how the SNPs influences the gene expression.
The functional effect of one of the studied SNPs is secondary to a change in the coding amino acid (rs1051266 missense mutation is the cause of SLC19A1-A80G amino acid polymorphism) although its exact effect on liver biology is still to be determined. RFC1 is a cell membrane folate transporter and, thanks to the work published just this month by Matherly and Hou (Nature, December 2022, Vol 612: 39-41) of the description of the RFC1 structure, we can confirm that changes in its amino acid composition can alter the protein conformation and/or the interaction with other molecules. We have included the reference of this recent work on page 51 of the amended manuscript.
Concerning the real effect of each SNP, it is the obvious continuation of this work, and hence, an in-progress extensive project.
- Is exessive folate suplementation in the media able to reduce the effects of SCL19A1 dowregulation. This would be of clinical interest.
This question is really interesting and somewhat complicated, as excessive folate can be excreted in the urine, it has been thought that excessive supplementation is not dangerous to one’s health. However, both low and high levels of folic acid similarly impact global DNA methylation, cytome biomarkers, DNA damage induced by oxidative stress, and DNA base excision repair gene expression (reference doi: 10.3390/cimb44040097 and comment added on page 52 of the reviewed manuscript). Furthermore, studies have found a link between high folate status in mothers and insulin resistance and adiposity in children (reference doi: 10.1007/s00125-007-0793-y and comment added on page 52 of the reviewed manuscript).

Round 2
Reviewer 1 Report
Thanks for taking into consideration my comments. Congratulations to all authors for this interesting piece of work.
Author Response
Thanks a lot for your help.
Best,
Ana M. Aransay